# HEMA-Lysine-Based Cryogels for Highly Selective Heparin Neutralization

**DOI:** 10.3390/ijms25126503

**Published:** 2024-06-13

**Authors:** Tommaso Mecca, Fabiola Spitaleri, Rita La Spina, Sabrina Gioria, Valentina Giglio, Francesca Cunsolo

**Affiliations:** 1CNR-Institute of Biomolecular Chemistry, Via Paolo Gaifami 18, 95126 Catania, Italy; tommaso.mecca@icb.cnr.it (T.M.); v.giglio@icb.cnr.it (V.G.); 2MEDIVIS, Via Carnazza 34C, 95030 Tremestieri Etneo, Italy; fabiola.spitaleri@medivis.it; 3Joint Research Centre (JRC), European Commission (EC), 2440 Geel, Belgium; rita.la-spina@ec.europa.eu (R.L.S.); sabrina.gioria@ec.europa.eu (S.G.)

**Keywords:** cryogels, HEMA, anticoagulation, heparin neutralization, filtration, medical device, dialysis

## Abstract

Unfractionated heparin (UFH) and its low-molecular-weight fragments (LMWH) are widely used as anticoagulants for surgical procedures and extracorporeal blood purification therapies such as cardiovascular surgery and dialysis. The anticoagulant effect of heparin is essential for the optimal execution of extracorporeal blood circulation. However, at the end of these procedures, to avoid the risk of bleeding, it is necessary to neutralize it. Currently, the only antidote for heparin neutralization is protamine sulphate, a highly basic protein which constitutes a further source of serious side events and is ineffective in neutralizing LMWH. Furthermore, dialysis patients, due to the routine administration of heparin, often experience serious adverse effects, among which HIT (heparin-induced thrombocytopenia) is one of the most severe. For this reason, the finding of new heparin antagonists or alternative methods for heparin removal from blood is of great interest. Here, we describe the synthesis and characterization of a set of biocompatible macroporous cryogels based on poly(2-hydroxyethyl methacrylate) (pHEMA) and *L*-lysine with strong filtering capability and remarkable neutralization performance with regard to UFH and LMWH. These properties could enable the design and creation of a filtering device to rapidly reverse heparin, protecting patients from the harmful consequences of the anticoagulant.

## 1. Introduction

Heparin is a polysaccharide mixture of different-length chains composed of repeated disaccharide units of 1→4 linked sulfated iduronic acid and sulfated glucosamine residues. Sulfo- and carboxyl groups displayed at defined intervals and orientations along the flexible polysaccharide backbone provide the highest negative charge density of any known biological macromolecule. Unfractionated heparin (UFH) and its low-molecular weight-fragments (LMWHs) are the most widely used anticoagulants in surgical practice and extracorporeal therapies such as heart–lung oxygenation and kidney dialysis [1]. In all of these cases, the anticoagulant effect of the administered heparin must be controlled and, if necessary, neutralized to prevent fatal bleeding events. Dialyzed people routinely use heparin during each dialysis session, several times a week and without any neutralization, so the accumulation of unmetabolized drugs often causes the onset of severe complications like heparin-induced thrombocytopenia (HIT) [2], osteoporosis [3], and other effects on metabolism such as hyperlipidemia. To date, protamine sulfate, a low-molecular-weight arginine-rich protein, is the only approved antidote to neutralize UFH [4]. This neutralization occurs through multivalent electrostatic interactions between the positively charged arginine residues of protamine and the negative sulfate and carboxylate residues of heparin [5]. However, the use of protamine is not very effective in neutralizing LMWHs [6], and it may cause side effects such as hypotension, bradycardia, thrombocytopenia, leukopenia, and anaphylactic shock itself [7,8]. For all these reasons, the development of antidotes able to selectively reverse heparin’s anticoagulant effect may have interesting therapeutic applications [5,9,10,11].

Therefore, peptides [12,13,14], engineered proteins [15], polycationic macrocycles [16,17] and particles [18], cationic polymers [19,20,21], porous cationic polymers [22], and dendritic polymer-based universal heparin reversal agent (UHRA) [23,24,25,26] have been proposed. During recent decades, functional porous materials with a well-defined porosity, a large surface area, and specific functional groups, due to their standout adsorbent properties, have been suggested for environmental treatment [27], energy storage [28], and blood purification [29]. In this scenario, cryogels, a class of polymeric materials formed below the solvent’s freezing point, in which the ice crystals act as porogens during the polymerization process, have aroused significant interest [30]. As ice-templated macropores, with diameters in the tens of microns, allow viscous fluids to pass through them more easily, cryogels have also been usefully applied in cell and protein separation [31,32].

In a previous paper, we described an *L*-lysine poly(2-hydroxyethyl methacrylate) (Old-pHEMA-lys) cryogel obtained by means of esterification, in a heterogeneous phase, between the primary alcoholic function of a monolithic pHEMA cryogel and Boc-Lys(Boc)-OH [33]. The material was characterized and the adsorption properties in a buffer solution and fresh human plasma towards UFH were evaluated. It also showed good blood compatibility, as indicated by the negligible uptake of albumin, antithrombin III, and total proteins, and, if properly designed, would enable the development of a heparin neutralization device which, when integrated into an extracorporeal blood circulation machine, would make medical practices safer for patients.

However, despite its interesting properties, this material presents some limitations due to the adopted functionalization procedure that provides only a single material with about 50–55% bonded lysine, low reproducibility, and inhomogeneous distribution of the active lysine residues. To overcome this problem, a new synthetic approach is developed her that allows for obtaining a set of cryogels with different and predictable compositions, tunable activity towards heparin neutralization, and homogeneous composition. Due to the potential applicative relapses, some of the reported results are previously disclosed in a patent declared below.

## 2. Results

### 2.1. Synthesis of L-Lysine Poly(2-hydroxyethyl methacrylate) Cryogels (pHEMA-lys)

Starting from 2-hydroxyethyl methacrylate (HEMA), via the esterification of the alcoholic function with Boc-Lys (Boc)-OH, followed by trifluoroacetic acid deprotection of the amine groups, the monomer HEMA-lys was obtained (Figure 1).

The copolymerization between HEMA and HEMA-lys in a wide range of molar ratios, in the presence of methylene-bis-acrylamide (MBAA) (Figure 2), at a temperature of −14 °C, allows for the realization of cryogels with tunable filtering materials and different neutralizing performances with regard to UFH and LMWH. Exploiting this synthetic procedure, three different materials, pHEMA-lys25, pHEMA-lys50, and pHEMA-lys75, at proportions of 25%, 50%, and 75% in the HEMA-lys composition, respectively, were synthesized and characterized. Any attempts to synthesize a homopolymer composed of 100% HEMA-lys failed due to the formation of a non-porous gel-like structure during the cryostructuration process. The three obtained materials were characterized by IR, TG, and SEM analysis together with the swelling behavior and porosity. Moreover, using ^1^H-NMR titration in the presence of an internal standard, their neutralization properties with regard to UFH and LMWH, in comparison with the previously described material (Old-pHEMA-lys), were studied.

### 2.2. Infrared Analysis

The FT-IR spectra (Figure 1a) show the progression of a signal at 1646 cm^−1^, possibly assignable to the bending vibrations of salified primary amine, according to the rise of lysine pendants for the three materials at increasing concentrations of HEMA-lys monomer.

### 2.3. Thermogravimetric Analysis

The thermal stability of the three samples was analyzed by TGA. Analysis of the overlayed thermograms for the three materials (Figure 1b) showed a similar profile with a degradation range from 250 °C to 450 °C, a maximum rate of decomposition between 426 °C and 434 °C, and a fixed residual between 1% and 3%.

### 2.4. Morphological Analysis

The scanning electron microscopy (SEM) analysis highlighted a marked difference in the morphology of the materials due to the different ratio of HEMA-lys monomer used in the synthesis (Figure 2). As the cryopolymerization reaction occurs at a slightly basic pH (7.5–8) and the lysine pendants are positively charged for more than 70%, several water molecules are involved in a strong hydrogen bond network depending on lysine pendant concentration. When freezing aqueous solutions, hydrogen-bonded water remains unfrozen at T < −15 °C, so the amounts of water in the different states (free, weakly, and strongly bonded) deeply affect the morphology of the materials in terms of pore dimensions and thickness of the walls [34]. Thus, when the concentration of polar pendants is low, more water crystals are formed, and a macroporous, interconnected network with thin pore walls characterizes the resulting material. As the concentration of the polar pendants increases, more thawed water remains, and the material turns into a gel-like structure with small, less connected pores and thick walls.

### 2.5. Swelling and Porosity

The equilibrium swelling degree was determined via gravimetric analyses by measuring the mass of water uptake by the sample in the form of trifluoroacetate salt at a temperature of 25 °C versus the mass of the dry sample. The results of the swelling experiments are summarized in Figure 3 and are in line with the morphological differences between the three materials evidenced by SEM images. Generally, the increase in a sample’s volume after water adsorption depends mainly on the amount of water adsorbed by the polymeric material that composes the walls of the cryogel structure. In the three synthesized samples, the amount of “gel-like” phase formed during the cryostructuration process, which grows with the lysine content, seems to have a key role in determining their swelling properties. In fact, this “gel-like” phase, during the drying process, shrinks and collapses permanently in a non-porous solid state, as evidenced by the shrinking of the whole sample, and water is no longer able to return the sample to its original state. As a consequence, the swelling of the three different materials decreases when increasing the lysine content.

The porosity percentage of dried samples was calculated as the ratio between the volume of adsorbed cyclohexane versus the total volume of samples. Cyclohexane is a solvent that is not adsorbed by the polymeric matrix, and, through capillarity, it fills only the pores of the cryogel. The results are summarized in Figure 3 and are consistent with the SEM analysis and swelling tests. The porosity decreases when increasing the lysine content in the polymeric material, reflecting the non-frozen water content during the cryopolymerization process: the higher the amount of non-frozen water, the higher the wall thickness, and consequently the pore volume determined by the frozen water during cryopolymerization is smaller.

### 2.6. Heparins Adsorption Tests by ^1^H-NMR Spectroscopy

The three cryogels, pHEMA-lys25, pHEMA-lys50, and pHEMA-lys75, were investigated regarding their capacity in sequestering UFH and LMWH in comparison with the material previously obtained by means of esterification in the heterogeneous phase (Old-pHEMA-lys) [33]. Quantitative titrations were performed by means of ^1^H-NMR spectroscopy in deuterated water, using tert-butyl alcohol as the internal standard, monitoring the area of the characteristic heparin signal at *δ* = 1.95 that linearly grows with the additions, thus indicating saturation of the material. The results are summarized in Figure 4, where the total amounts of UFH or LMWH neutralized by the materials are reported. These experiments highlight a significant increase in the new materials’ sequestration capacity with regard to UFH as well as LMWH compared to the Old-pHEMA-lys material, whose percentage of functionalization was established at about 50–55%. However, it should be noted that the new material pHEMA-lys50, whose activity should be similar to that of Old-pHEMA-lys, allows the sequestering of a considerably greater quantity of heparin, probably due to the homogeneous distribution of the lysine in the material.

### 2.7. Homogeneity of pHEMA-lys50 versus Old-pHEMA-lys

The inhomogeneous distribution of lysine pendants in the Old-pHEMA-lys cryogel was verified by UFH titrations performed on three samples obtained from three slices of the old material (central, intermediate, and outer) compared to the new pHEMA-lys50 material. The histograms in Figure 5 show how the slice obtained from the outermost area of the Old-pHEMA-lys cryogel appears to be significantly more active in the neutralization of heparin than the intermediate and central parts of the analyzed samples. Meanwhile, the comparison with the three slices of pHEMA-lys50 unequivocally shows the homogeneous composition of the material obtained with the proposed synthetic procedure.

### 2.8. Hemocompatibility Tests

With this promising activity in the neutralization of heparins, to verify whether the material could come into contact with blood, hemocompatibility tests such as hemolysis and hematology, as well as complement C3, were preliminarily performed using a sample of pHEMA-lys25 compared to plain pHEMA [35].

#### 2.8.1. Induced Hemolysis by the Cryogel

To evaluate if the materials, in contact with the blood, may cause damage to red blood cells and produce increased levels of free plasma hemoglobin, tests were performed by mixing the heparinized blood of two donors with samples of the two materials under observation. The following table shows the induction of hemolysis by the cryogels pHEMA and pHEMA-lys25 over 24 h and represents the average of three independent experiments. The amount of hemoglobin was assessed using colorimetric Drabkin’s reagent at 540 nm. To calculate the degree of hemolysis for test compounds, positive (Triton™ X-100) and negative (4% polyethylene glycol solution, average mol wt 8.000) controls were included to normalize the results. The results shown in Table 1 suggest that the pHEMA and pHEMA-lys25 materials exert favorable hemolytic activity lower than 1%.

#### 2.8.2. Count of White Blood Cells (WBCs) and Platelets (PLTs)

A count of WBCs and PLTs was performed to complete the evaluation of the materials regarding blood contact. The data shown in Table 2 highlight that the variation of WBCs in pHEMA and pHEMA-lys25 samples is 2% and 4%, respectively, whereas the variation of PLTs is 4% and 10%, respectively. These results are still acceptable from the point of view of the materials’ hemocompatibility.

#### 2.8.3. Complement C3 Human Test

The complement activation determines the potential immune response caused by introducing a foreign material into the body that can lead to adverse effects such as inflammation and tissue damage. The test was performed using a complement C3 Human ELISA kit. The following results, presented in Table 3, recorded after 2 h and 24 h for serum in contact with the materials, show that complement C3 activation does not take place, supporting, as a first estimation, the material hemocompatibility hypothesis.

## 3. Discussion

In the present work, we describe the progress and improvement of the cryopolymerization process that provides crosslinked polymeric materials, based on pHEMA-lys, in the form of a macroporous gel able to sequester the anticoagulant heparin (UFH) and its low-molecular-weight derivatives (LMWH) from aqueous solutions and biological fluids. The advantage of this process is the possibility of obtaining homogeneous materials whose functionalization percentage can be set at will, determining their composition and neutralization activity a priori. Moreover, the results of the NMR titration experiments (Figure 4), designed to assess the affinity for UFH and LMWH of three materials with different compositions (25%, 50%, and 75% pHEMA-lys) in comparison with Old-pHEMA-lys, whose percentage of functionalization is in the order of 50–55%, indicated a significant increase in the sequestering capacity of the new materials. The greater the amount of HEMA-lys present in the cryogel, the greater the amount of heparin sequestered per gram of polymer. Using previous cryogel synthetic techniques, it was only possible to obtain a cryogel with a nominal functionalization of 50–55% (Old-pHEMA-lys). However, it should be noted that the cryogel at 50% functionalization with lysine sequesters a quantity of heparin that is considerably greater than the amount sequestered by the Old-pHEMA-lys cryogel in all cases.

The lower activity and the lack of homogeneity of the old material can be explained by the previously adopted synthetic protocol. In this case, a pHEMA sample already structured in the solid state as a cryogel is functionalized in the heterogeneous phase in the presence of Boc-Lys(Boc)-OH using N,N′-diisopropylcarbodiimide (DIC) as a coupling agent for the esterification reaction. As the reaction proceeds, DIC form a urea by-product that is poorly soluble in DMF which remains deposited in the outer part of the sample, slowing down the access of reagents to the innermost parts of the material. At the end of the procedure, washing the final material in ethanol dissolves the urea by-product, providing a porous but unevenly functionalized material. Due to this phenomenon, the external part of the material is more functionalized than the internal part. To partially overcome this problem, the end parts of the cryogels thus prepared are discarded for the purposes of performing the reported experiments. Moreover, during the synthesis, owing to the swelling effect of the organic solvent, the solid polymer increases its volume, resulting in pore enlargement. At the end of the reaction, when washed with aqueous/alcoholic solvents, the polymer shrinks and returns to its original shape. This process markedly affects the material’s porous structure and some amino groups arranged in the inner areas remain trapped inside and are no longer available for complexation with heparin.

In contrast, the process here described is a homogeneous synthesis process in which the reagents HEMA and HEMA-lys, both soluble in water, react together to form the cryogel in its final structure. In this case, the assembling of the material is driven by the electrostatic equilibrium between the solvated charged lysine groups present in the HEMA-lys monomer. Water, therefore, doubly influences the structuring of the material, both in the formation of the pores because of the ice crystals formed during the polymerization process as well as in the ordered arrangement of the charged groups of the lysine pendants. At the end of the process, the new material has a more homogeneous composition compared to the Old-pHEMA-lys material, with the charged amino groups arranged towards the outside in contact with the aqueous phase. Consequently, the activity of the material is also improved thanks to the greater availability of the lysine amino groups for complexation with heparin.

## 4. Materials and Methods

### 4.1. Materials

Boc-Lys(Boc)-OH dicyclohexylammonium salt, 4-dimethylaminopyridine (DMAP), N,N′-dicycloexylcarbodiimide (DCC), 2-hydroxyethyl methacrylate (HEMA), N,N′-methylene-bisacrylamide (MBAA), ammonium persulfate (APS), tetra-methyl-ethylene-diamine (TEMED), unfractionated heparin (UFH) sodium salt from porcine intestinal mucosa (H-3149, 213 units mg^−1^), low-molecular-weight heparin (LMWH) sodium salt from porcine intestinal mucosa (H-3400, <60 U/mg), and Drabkin′s reagent were purchased from Sigma-Aldrich (St. Louis, MO, USA). The C3 Human ELISA kit (ab108822) was purchased from Abcam (Cambridge, UK).

### 4.2. Synthesis of HEMA-lys Monomer

As the lysine reagent was in the form of dicyclohexylamine salt (Boc-Lys(Boc)-OH DCHA), the first step involved extraction using a citric acid aqueous solution (10% *w*/*v*). Specifically, 2 g of Boc-Lys(Boc)-OH DCHA dissolved in 70 mL of dichloromethane was extracted using 70 mL of a citric acid aqueous solution (×3 times), then the organic layer was washed with water, dried on anhydrous sodium sulfate, and evaporated. In this way, 1.3 g of Boc-Lys(Boc)-OH (98% yield) was thus obtained. The Boc-protected lysine (1.203 g, 3.473 mmol) was mixed with DMAP (43.02 mg, 0.347 mmol) and DCC (788 mg, 3.82 mmol) and the mixture was dissolved in 15 mL of dry CH_2_CI_2_ and 2 mL of dry DMF under an inert atmosphere. After complete dissolution, the temperature of the reaction mixture was lowered to 0 °C and 464 μL of HEMA (498 mg, 3.83 mmol) was added. The reaction was thus continued under stirring for 30 min at 0 °C, and subsequently at room temperature for 24 h. The mixture was then filtered, the solvent was eliminated by means of evaporation, and the resulting residue was purified by means of chromatography (SiO_2_, Hexane/Ethyl Acetate: 70/30). Small amounts of radical inhibitor (p-methoxyphenol) were added to the eluting solvent. In this way, 1.35 g of pure HEMA-Boc-Lys(Boc) (Yield: 80%) was obtained. The next synthetic step involved removing the tert-butoxycarbonyl groups to restore the free amino groups. This reaction was performed by solubilizing the HEMA-Boc-Lys(Boc) compound in neat trifluoroacetic acid and maintaining it under stirring at room temperature for about 90 min. At the end of the reaction, the system was brought to dryness using a rotary evaporator, dissolved in ethanol, and dried again three times. In this way, 1.36 g of pure HEMA-lys (Yield: 99%) was thus obtained. ^1^H NMR (400 MHz, MeOD): *δ* 1.45–1.63 (m, 2H), 1.70 (p, 2H, *J* = 7.8 Hz), 1.85–2.03 (m, 2H), 1.94 (dd, 3H, *J*_1_ = 1.6 Hz, *J*_2_ = 1.0 Hz), 2.93 (t, 2H, *J* = 7.8 Hz), 4.08 (t, 2H, *J* = 6.5 Hz), 4.40–4.51 (m, 3H), 4.55–4.63 (m, 1H), 5.67 (p, 1H, *J* = 1.6 Hz), and 6.12 (dq, 1H, *J*_1_ = 1.6 Hz, *J*_2_ = 1.0 Hz).

### 4.3. Synthesis of pHEMA-lys Cryogel

The general synthesis of cryogels involved a sum of the monomer concentration equal to 0.5 M and a monomer/crosslinker ratio of 6:1.

HEMA-lys and HEMA in various mutual ratios were dissolved in 159 μL of a 0.162 M MBAA solution in water. The temperature of the mixture was lower to 0 °C, then small portions of 1.5 M NaOH solution were added until a pH of about 7.5 was reached, and subsequently the final volume was corrected by the addition of H_2_O (final volume = 310 μL, including the addition of APS and TEMED; sum of monomers concentration = 0.5 M). Two aqueous solutions of APS and TEMED, both at 10% *w*/*v*, were prepared. To the monomer solution, 6.2 μL of the APS solution and subsequently 6.2 μL of the TEMED solution were added under stirring. The mixture thus obtained was poured into a closed cylindrical glass reactor (inner diameter 7 mm) and cooled by means of immersion in a refrigerated bath at a temperature of −14 °C for 24 h. The material obtained after solvent thawing was washed with water and ethanol, dried under a nitrogen flow and then under vacuum. Table 4 shows the ratio of HEMA-lys/HEMA (mmol) for each prepared sample in a total volume of 310 μL.

### 4.4. Infrared Analysis

ATR FT-IR spectra were acquired in the 4000–400 cm^−1^ region on the PerkinElmer spectrum using two FT-IR spectrometers equipped with an internal reflection crystal of zinc selenide (ZnSe). Thin slides of each sample, weighing about 2 mg, were prepared for analysis. Each spectrum was the result of the average of 32 scans at a resolution of 4 cm^−1^.

### 4.5. Thermogravimetric Analysis

The thermal stability of the full set of samples was established through a PerkinElmer TGA 8000 apparatus. The analyses were carried out on samples of about 3 mg under a nitrogen flow (60 mL/min), with a heating ramp of 10 °C/min, in the temperature range from 50 °C to 700 °C. Data were acquired and analyzed through Pyris Software V.13.3.3.0032, supplied by the producer.

### 4.6. Morphological Analysis

The morphology of each sample was evaluated through a scanning electron microscopy apparatus (SEM), Phenom Desktop G5, purchased from Phenomenex (Torrance, CA, USA). Images and data were explored using the Phenom-World software ProSuite 2.9.0.0 bundle package, Eindhoven, The Netherlands.

### 4.7. Swelling Test

The water-absorbing ability of each sample was tested by performing a swelling test by weighing the dried specimen, in the form of trifluoroacetate salt, and then dipping it in degassed water for two hours at a temperature of 25 °C. The wet samples were moved onto the scale, a small amount of water was added to correct any water loss during sample collection and transport, and subsequently any excess of water was gently removed with absorbent paper. The weight measured after the dipping allowed us to calculate the swelling degree (*S*) using the following equation [36]:$$S=\frac{m_{w}-m_{d}}{m_{d}}\;\;\;\;\;\;\;\;\;\;\;\;\;\;\;\;\;\;\;\;\;$$
where *m_w_* represents the weight of the wet sample after solvent uptake and *m_d_* is the weight of dried sample.

### 4.8. Porosity

The porosity of the dried samples was calculated as a percentage by measuring the adsorbed volume of cyclohexane versus the total volume of each sample according to Archimedes’ principle with the help of an adapted gravity bottle and determined using the following equation [36,37]:$$P\%=\frac{V_{pores}}{V_{sample}}\times 100=\frac{m_{w}-m_{d}}{m_{1}-m_{2}+m_{w}}\times 100\;\;\;\;\;\;\;\;\;\;\;\;\;\;\;$$
where *m_w_* is the mass of the cyclohexane-saturated cryogel, *m_d_* is the mass of the dried cryogel, *m*_1_ is the mass of the gravity bottle filled with cyclohexane, and *m*_2_ is the mass of the gravity bottle containing both cyclohexane and the cryogel. The samples were immersed in cyclohexane under reduced pressure before weighing in order to remove the residual gas trapped in the cryogels. This operation was repeated before their transfer into the gravity bottle.

### 4.9. Adsorption Capacity (NMR Titrations)

The ability of each material to adsorb heparins was estimated via ^1^H-NMR spectroscopy titrations. Small aliquots of UFH or LMWH solution (20 mg/mL) were added to an NMR tube containing 10 mg of each material suspended in D_2_O (600 μL), with *tert*-butyl alcohol (0.5 μL) as the internal standard for the calculation of the areas. The titration was continued until the area of one of the characteristic heparin signals at δ = 1.95 linearly grew with the additions, thus indicating saturation of the material. All titration experiments were repeated three times using samples from three different cryopolymerization syntheses. By plotting the area of the signal (with respect to the internal standard) against the amount of heparin added (in mg), it was possible to create a graph (see Appendix A) whose intercept on the X axis of the straight line that best approximates the linear trend of the heparin signal indicated the total amount of heparin sequestered by the cryogel sample.

### 4.10. Samples Preparation for Homogeneity Tests

The homogeneity of the two materials Old-pHEMA-lys and pHEMA-lys50 was compared by cutting 1 cm long cylindrical samples of both materials into 5 slices of approximately 2 mm each, as shown in Figure 5. NMR titration towards UF heparin was performed for each slice with the methodology already described in Section 4.9. Tests were performed in triplicate using slices of samples from different synthesis batches.

### 4.11. Hematology Test

The induced hemolysis test was performed using colorimetric Drabkin’s reagent at 540 nm using a UV-vis spectrophotometers Agilent 8453. For the count of white blood cells (WBCs) and platelets (PLTs), blood was collected in an EDTA-stabilized tube and diluted to 1:1 with a physiological solution. Then, 3 mL of the diluted blood solution was placed in contact with 11 mg of pHEMA and pHEMA-lys. The test was carried out using the Advia 120 hematology system, Siemens.

### 4.12. Complement C3 Human Test

The complement C3 human test was performed using a complement C3 Human ELISA kit (ab108822) purchased from Abcam. In this test, 20 mg of dry material was used in 1.8 mL of a 1:20 serum–physiological solution.

## 5. Conclusions

We have developed a new method for achieving cryostructurated materials with excellent filtering and capturing properties for heparin and its low-molecular-weight homologues, quantitatively evaluating their neutralization capabilities. These materials were thoroughly examined for their physical/chemical properties, morphology, and hemocompatibility. Using these materials in designing filtering devices for heparin neutralization could improve the safety of surgical and therapeutic practices.

Furthermore, thanks to the excellent filtering properties of these materials, preliminary tests have been carried out on their ability to absorb pollutants from water. These tests have shown a high affinity towards bisphenol A, a monomer used in the manufacture of either polycarbonate plastic or epoxy resins considered extremely harmful to humans and the environment. The results were promising and suggest that further research could open up new avenues for these materials’ application in environmental remediation as well.

## 6. Patents

Some of the results discussed in this work have been reported in the patent: Cryogel for the removal of heparins and heparinoids from aqueous solutions, physiological solutions and biological fluids, preparation process and uses thereof. PCT/IB2017/050712

## Data Availability

The data presented in this study are openly available in this article.

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
