# Peer review of "HEMA-Lysine-Based Cryogels for Highly Selective Heparin Neutralization"

_ijms, 2024, doi:10.3390/ijms25126503_

Round 1

Reviewer 1 Report

Comments and Suggestions for Authors

Mecca et al. described the synthesis and characterization of a series of poly(2-hydroxyethyl methacrylate) (pHEMA) and L-lysine hydride macroporous cryogels (pHEMA-lys) for heparin neutralization. While the authors thoroughly characterized cryogel morphology, functional groups, and swelling, additional experiments are needed to support the stated aims regarding reproducibility, homogeneous lysine distribution.

The Introduction indicates “However, despite its interesting properties, this material presents some limitations due to the adopted functionalization procedure that provides only a single material with about 50-55% bonded lysine, low reproducibility, and inhomogeneous distribution of the  active lysine residues. To overcome this problem, it has been developed a new synthetic approach that allows obtaining a set of cryogels with different and predictable compositions, tunable activity towards heparin neutralization, and homogeneous composition.” Thus, data should be provided demonstrating: 1) reproducibility; 2) homogeneous composition. Those two are key information should be delivered by the work.

Additionally, the differences between the proposed pHEMA-lys synthesis method and prior pHEMA-lys protocols should be discussed.

Overall, the fundamentals of the cryogel synthesis and characterization are sound. However, additional experiments directly comparing the reproducibility, homogeneity of the new pHEMA-lys cryogels to old formulations would strengthen the conclusions and support the stated advances of this improved synthetic approach.

Minor comments:

1. Add y-axis descriptions in Figure 3 plots.

2. Include statistical analysis between experimental groups.

3. Revise language and grammar to improve clarity and readability.

Author Response

Thank you for reviewing our work, your advice was useful and improved the result of our work.
Attached you will find detailed answers to the points you highlighted
Best regards

Reviewer 2 Report

Comments and Suggestions for Authors

The study and results are well done and worth publishing but some more explanation and clarification are needed.

Why lysine-based monomers were boc-deprotected before polymerization?

During polymerization is it any difference in polymerization rate with increasing of temperature, since APS/TEMED usually most active at 22 C.

After gel formation are that gels hold the form or mostly low crosslinked jelly-like structures? Maybe adding some photographs of materials?

During swelling experiment: was gels neutralized and measured at neutral pH, since lysine-based polymers have charged amino groups and it would effect on swelling. What was temperature of swelling experiments? Please add all experimental info to the text.

What means: ” homogeneous distribution of the lysine in the material”? Is it was calculated amount of lysine in final polymers? What about constant of copolymerization. Did you measured rate of polymerization?

Is it any effect of porosity on heparin absorption? Did you use fine dispersion of materials or chanks of gels, is it any effect of sizes of materials on selective heparin neutralization?

Author Response

(The authors gave the same response as above.)

Round 2

Reviewer 1 Report

Comments and Suggestions for Authors

I appreciate the efforts the authors have put into addressing my questions and making substantial changes to enhance the quality of their work. The work could be accepted for publication in its current form.